# Comparing Efficacy of Erlotinib and Bevacizumab Combination with Erlotinib Monotherapy in Patients with Advanced Non-Small Cell Lung Cancer (NSCLC): A Systematic Review and Meta-Analysis

**DOI:** 10.3390/diseases11040146

**Published:** 2023-10-23

**Authors:** Prashant Sakharkar, Sonali Kurup

**Affiliations:** 1Department of Clinical and Administrative Sciences, College of Science, Health and Pharmacy, Roosevelt University, Schaumburg, IL 60173, USA; 2Department of Pharmaceutical Sciences, College of Pharmacy, Ferris State University, Big Rapids, MI 49307, USA; sonalikurup@ferris.edu

**Keywords:** erlotinib, bevacizumab, non-small cell lung cancer, NSCLC, epidermal growth factor receptor tyrosine kinase inhibitor, EGFR-TKI, vascular endothelial growth factor receptor tyrosine kinase inhibitor, VEGFR-TKI

## Abstract

The objective of this systematic review and meta-analysis was to assess and contrast the efficacy and safety of combining erlotinib and bevacizumab with erlotinib alone in the treatment of patients with advanced non-small cell lung cancer (NSCLC). The authors searched databases such as PubMed, Medline, Scopus, and Cochrane Central Register of Controlled Trials for randomized control trials (RCTs) comparing erlotinib plus bevacizumab with erlotinib in NSCLC patients. The overall survival (OS), progression-free survival (PFS), objective response rate (ORR), and adverse events (AEs) were the outcomes of interest. The pooled hazard ratio (HR) and relative risk (RR) were estimated utilizing both fixed- and random-effect models. Methodological quality of the included studies was assessed using the Cochrane Risk of Bias tool. Nine studies comprising 1698 patients with NSCLC were included in this meta-analysis, of whom 850 were treated with erlotinib plus bevacizumab, and 848 with erlotinib. The erlotinib plus bevacizumab combination significantly prolonged PFS (HR, 0.62, 95% CI: 0.56, 0.70, *p* < 0.00001) but did not show any significant improvement in OS (HR, 0.95; 95% CI: 0.83, 1.07, *p* = 0.39) and ORR (HR, 1.10; 95% CI: 0.98, 1.24, *p* = 0.09). Increased risks of hypertension (RR, 5.15; 95% CI: 3.59, 7.39; *p* < 0.00001), proteinuria (RR, 10.54; 95% CI: 3.80, 29.20; *p* < 0.00001) and grade 3 and higher AEs (RR, 2.09; 95% CI: 1.47, 2.97; *p* < 0.00001) were observed with the erlotinib-plus-bevacizumab combination compared to erlotinib monotherapy. On subgroup analyses, the erlotinib plus bevacizumab combination improved PFS only. Combining erlotinib and bevacizumab has been shown to improve PFS in advanced NSCLC patients but did not show any significant OS and ORR benefits. Furthermore, risks of hypertension, proteinuria, and grade 3 or higher AEs were greater with the erlotinib-and-bevacizumab combination.

## 1. Introduction

Lung cancer is the second most common cancer in both men and women in the United States [1]. Non-small cell lung cancer (NSCLC) comprises approx. 85% of all cases. It is the most common among all types of lung cancer [1]. Despite recent advancements in treatment, patients with advanced NSCLC have a poor prognosis, with a high rate of recurrence and metastasis, and a five-year survival rate of only 5.5% [2]. The conventional approach for treating advanced NSCLC has been platinum-based chemotherapy, which has significant side effects and provides limited benefits in terms of overall survival [3,4]. The immune checkpoint inhibitor-based treatment, not a cytotoxic chemotherapy alone, is the current standard of care in metastatic NSCLC without a driver mutation or fusion.

Erlotinib, a tyrosine kinase inhibitor (TKI), focuses on inhibiting the epidermal growth factor receptor (EGFR). Its effectiveness in enhancing both progression-free survival (PFS) and overall survival (OS) among individuals with EGFR + NSCLC has been demonstrated in studies [5,6,7]. The occurrence of EGFR mutations in metastatic NSCLC is around 10–20% in Europe and 40–60% in Asia [8], underscoring the potential benefits of utilizing erlotinib in this specific group.

Similarly, Bevacizumab is a monoclonal antibody that targets vascular endothelial growth factor (VEGF), a key regulator of angiogenesis. Bevacizumab, when combined with platinum-based chemotherapy, has been shown to improve OS [9,10] in advanced NSCLC patients. In preclinical studies, the combination of erlotinib and bevacizumab has demonstrated the synergistic inhibition of tumor growth [11], invasion, and metastasis in NSCLC [12]. In Caucasian patients with EGFR-mutated NSCLCs in one phase II study [13], erlotinib and bevacizumab as a first-line treatment showed improved PFS. However, this study did not include a control group for comparison, thus making these results inconclusive. The erlotinib and bevacizumab combination in one phase III clinical study showed greater PFS in unselected populations of patients with NSCLC [14]. On the other hand, studies that compared Osimertinib plus bevacizumab against Osimertinib monotherapy failed to demonstrate improved PFS in patients with NSCLC harboring EGFR mutations [15,16,17].

Previous clinical trials comparing the erlotinib-plus-bevacizumab combination with erlotinib monotherapy have yielded conflicting results, with some demonstrating benefits in terms of PFS, OS or the objective response rate (ORR), and others showing no significant differences between the two treatment regimens. Additionally, some studies were underpowered [18,19]; some failed to achieve treatment outcomes [19], or suffered from reporting and publication bias [20]. Previous meta-analyses [21,22] have also investigated the effects of EGFR-TKIs in combination with VEGFR-TKI compared to platinum-based chemotherapy or EGFR-TKI monotherapy. However, a previously conducted meta-analysis on this topic only included six RCTs, primarily from Asia [23].

Using a PICO framework (Patient, Intervention, Comparator, Outcomes), we conducted a meta-analysis to investigate whether the combination of erlotinib and bevacizumab is associated with improved OS, PFS, and ORR compared to erlotinib monotherapy in NSCLC patients. We included eligible trials that reported on the outcomes of interest as well as adverse events (AEs). Additionally, we performed subgroup analyses to assess the impact of geographic regions, mainly Asia, the United States, and Europe combined, on the treatment outcomes.

## 2. Materials and Methods

### 2.1. Data Source and Literature Search

The systematic review followed the recommendations of the Preferred Reporting Items for Systematic Reviews and Meta-analyses (PRISMA) [24]. The protocol has not been registered. The PRISMA checklist ensures the methodological rigor of systematic reviews and meta-analysis (Appendix A). To identify relevant articles, we systematically searched several databases, including PubMed, Medline, Scopus, the Cochrane Central Register of Controlled Trials, American Society of Clinical Oncology (ASCO), International Association for the Study of Lung Cancer (IASLC), European Society of Medical Oncology (ESMO), and clinicaltrials.gov. The databases were most recently searched in March 2023, and the following related keywords, MeSH terms, and their combinations were used: NSCLC, non-small cell lung cancer, lung cancer, erlotinib, erlotinib hydrochloride, bevacizumab and survival (Appendix A). Additionally, we searched the bibliographies of relevant articles for additional publications. Two researchers independently performed this literature search (PS and SK). If multiple articles covered the same study population, we used the data from the most recent study. Any discrepancies discovered between the two reviewers were addressed through mutual agreement.

### 2.2. Inclusion and Exclusion Criteria

We only included randomized clinical trials that met the following conditions: (1) had cytologically or histologically confirmed advanced EGFR-mutant NSCLC patients; (2) featured comparison involving erlotinib-plus-bevacizumab with erlotinib as first or second line of treatment; and (3) reported one or more survival outcomes (OS, PFS, ORR). We did not include single-arm studies, case reports, or animal experimentation. In addition, systematic reviews, meta-analysis, and those published in languages other than English were also excluded.

### 2.3. Data Extraction and Quality Assessment

Two investigators (PS and SK) independently reviewed all the selected articles, performed data abstraction, and assessed the methodological quality of the studies. The data extracted from the studies included the first author, year of publication, region, study type, number of patients in each group, type of EGFR mutation, therapeutic regimen, survival outcomes, and information on adverse drug events (Appendix A). The methodological quality of the studies included in the investigation was evaluated using the Cochrane Risk of Bias tool [25], which assesses quality based on following seven domains: (1) generation of random sequences; (2) concealment of allocation; (3) masking of participants and staff; (4) masking of outcome evaluation; (5) handling of incomplete outcome data; (6) prevention of selective reporting; and (7) identification of other potential biases. In instances of disagreement, a consensus was reached.

### 2.4. Bottom of Form Statistical Analysis

The OS and PFS were used as the primary endpoints to evaluate the efficacy of the treatments. We estimated the pooled hazard ratios (HR) of OS and PFS with a 95% confidence interval (CI). The relative risk (RR) with a 95% CI was used to estimate the results of ORR and AEs. We performed subgroup analyses to compare the treatment effects according to geographic location, which included Asia, the United States, and Europe. The I^2^ statistic was utilized to assess the degree of heterogeneity, where I^2^ values less than 25% were defined as low, 25–50% as moderate, and greater than 50% as substantial heterogeneity [26]. First, we employed random-effects models to accommodate the inherent variability within and between studies, owing to the limited number of trials. If the identified heterogeneity proved insignificant, a fixed-effects model was employed [27]. We did not perform sensitivity analysis due to the small number of eligible studies and the fact that most of them were open-label. To assess publication bias, we utilized the funnel plot and conducted Egger’s test. Funnel plots visually present effect sizes plotted against their standard errors or precisions. A skewed funnel plot shows the presence of publication bias. This can be intuitively assessed by examining the funnel plot symmetry [28]. On the other hand, in Egger’s test, standardized effect sizes are regressed on their precision. The regression intercept of zero indicates the absence of publication bias [29]. All statistical computations were conducted using RevMan V.5.4 from the Cochrane Collaboration and STATA ver. 14. Statistical significance was defined as a *p*-value of <0.05.

## 3. Results

### 3.1. Results of the Literature Search

The study selection process is visually presented in Figure 1. Initially, 797 records were identified through the search. After removing duplicates, 305 records were excluded. Seventeen articles were excluded during screening for eligibility. This resulted in a final set of nine studies [30,31,32,33,34,35,36,37,38] that met the inclusion criteria and were included in the meta-analysis.

### 3.2. Characteristics of the Included Studies

Table 1 summarizes the characteristics of the nine included studies. The meta-analysis included a total of 1698 patients from these nine studies. The NEJ026 study [31,32] and the JO25567 study [35,36] had dual publications included in the analysis. The erlotinib-plus-bevacizumab group consisted of 850 NSCLC cases, while the erlotinib group included 848 cases. Among the nine included RCTs, five were phase-3 trials and four were phase-2 trials. The earliest study was published in 2011 and the most recent in 2023. The sample size in these studies ranged from 88 to 363 patients. The median age of the patients ranged from 58 to 67 years old, and the proportion of female patients ranged from 46% to 70%. Seven studies included NSCLC patients with an L858R or del19 EGFR mutation. Patients in all studies received an erlotinib-plus-bevacizumab combination as a first line of treatment, except for one study.

### 3.3. Risk of Bias and Quality Assessment

Figure 2A provides a summary of the risk of bias for each included study. The majority (8) of the studies were open-label. Only one study had adequate concealment of allocation. All studies were free of incomplete outcome data and other biases. The studies included in this meta-analysis were found to be of a moderate to high quality (Figure 2B).

### 3.4. Progression-Free Survival

Except for two studies, the selected studies reported a statistically significant improvement in PFS. The median PFS ranged from 3.4 to 17.9 months in the combination group compared to 1.7 to 13.5 months in the erlotinib monotherapy group (Table 2).

Our meta-analysis demonstrated a significant increase in PFS when comparing the combination of erlotinib and bevacizumab to the erlotinib monotherapy group (HR, 0.62, 95% CI: 0.56, 0.70, *p* < 0.00001) (Figure 3A).

In the subgroup analyses, PFS was significantly prolonged among patients receiving erlotinib plus bevacizumab (HR, 0.59; 95% CI: 0.50, 0.71; *p* < 0.00001) among studies conducted in Asia, more than the studies conducted in the US/Europe (HR, 0.64, 95% CI: 0.55, 0.75; *p* < 0.00001) (Figure 3B) compared to erlotinib monotherapy. No heterogeneity was observed across all the analyses. The test for subgroup differences indicated no statistically significant subgroup effect (*p* = 0.50). This suggests that the geographic region does not modify the effect of erlotinib-plus-bevacizumab combination compared to erlotinib monotherapy. It is interesting to note that the pooled effect estimates for both subgroups favored the erlotinib-plus-bevacizumab combination.

### 3.5. Overall Survival

All included studies reported no statistically significant improvement in OS. The median OS ranged between 9.3 to 50.7 months in the combination group compared to 9.2 to 50.6 months in the erlotinib monotherapy group (Table 2). The meta-analysis showed that the erlotinib-plus-bevacizumab combination did not significantly prolong OS compared to erlotinib monotherapy in the entire study population (HR, 0.95; 95% CI: 0.83–1.07, *p* = 0.39) (Figure 4A). Subgroup analyses also did not show a significant improvement in OS with the combination therapy (Figure 4B). No heterogeneity was observed across all the analyses. The geographic region did not significantly modify the effect of the combination therapy compared to monotherapy on the test for subgroup differences (*p* = 0.99).

### 3.6. Objective Response Rate

Except for two studies, the selected studies reported no statistically significant improvement in ORR. The median ORR ranged between 12.6% to 86.8% among patients in the combination group compared to 6.2% to 83.9% in the erlotinib monotherapy group (Table 2). The results indicated that the erlotinib-plus-bevacizumab combination did not significantly improve ORR compared to erlotinib monotherapy in the entire study population (HR, 1.10; 95% CI: 0.98, 1.24, *p* = 0.09) (Figure 5A). Subgroup analyses also did not show significant improvement in the ORR with the combination therapy (Figure 5A–C). Significant heterogeneity was observed across all analyses except for the subgroup analysis of studies from Asia. Geographic region had no effect on the treatment outcomes of the erlotinib-plus-bevacizumab combination compared to erlotinib monotherapy (*p* = 0.17).

### 3.7. Adverse Events

The AEs associated with the use of erlotinib plus bevacizumab were compared to those associated with erlotinib monotherapy. Only AEs of grade 3 and higher were analyzed (Table 3).

Values are reported as a number (%); ERL + BEV, erlotinib plus bevacizumab; ERL, erlotinib monotherapy; AE, adverse events; HTN, hypertension.

The results of the meta-analysis showed that the combination therapy was associated with a higher incidence of high-grade AEs compared to the monotherapy. Specifically, patients treated with erlotinib plus bevacizumab had significantly increased risks of hypertension and proteinuria. However, the risk of skin rash and diarrhea were comparable between the two groups (Table 4). Significant heterogeneity was observed among studies included for the analyses of AEs of grade 3 and higher and skin rash; however, analyses of other AEs showed no heterogeneity. The forest plots for all AEs are presented in the Appendix A.

### 3.8. Publication Bias

There was no evidence of apparent publication bias based on the assessment using a funnel plot and Egger’s test (*p* > 0.05).

## 4. Discussion

In this meta-analysis, we systematically reviewed nine randomized clinical trials, which included a total of 1698 patients, to assess the clinical value and safety of erlotinib-plus-bevacizumab combination therapy. Our meta-analyses included randomized clinical trials that had cytologically or histologically confirmed advanced EGFR-mutant NSCLC patients; patients who were either recurrent or refractory after standard first-line chemotherapy or chemoradiotherapy or received no first line treatment, older than 18 years of age, with ECGOG-PS scores ranging from 0 to 4, with or without brain metastasis, stage I to IV, with an EGFR mutation (either exon 19 deletion or L858R mutation); comparing erlotinib-plus-bevacizumab to only erlotinib as a first or second line of treatment; and reported one or more survival outcomes (OS, PFS, ORR). Patients in these studies were excluded if they had received EGFR inhibitors or VEGF receptor inhibitors previously, had cardiovascular diseases (myocardial infarction, unstable angina, congestive heart failure, symptomatic arrhythmia, substantial peripheral vascular disease, uncontrolled hypertension, etc.), coagulopathy, abnormal hematological and liver function tests, or used warfarin or equivalents.

The treatment efficacy of combination therapy was evaluated with OS and PFS as the primary end points, using pooled HRs with a 95% CI, whereas the ORR and AEs were assessed using the pooled RR with a 95% CI, using either fixed- or random-effect models. The I^2^ statistics were used to demonstrate the heterogeneity among the studies included. The small number of studies, and most of them being open label, limited the scope of sensitivity analysis. Publication bias was assessed using a funnel plot and Egger’s test. The findings of our meta-analysis suggested that a combination of erlotinib and bevacizumab was effective in prolonging PFS in EGFR-positive NSCLC patients.

Patients in the erlotinib-plus-bevacizumab combination group had a median PFS ranging from 3.4 to 17.9 months, compared to 1.7 to 13.5 months in the erlotinib group. The median OS ranged from 32.4 to 50.7 months in the combination group, compared to 22.8 to 50.6 months in the erlotinib group. The median ORR ranged from 12.6% to 86.8% in the combination group, compared to 6.2% to 83.9% in the erlotinib monotherapy group.

The greater PFS among the erlotinib and bevacizumab combination therapy group was compared to the erlotinib monotherapy group, observed in a study by Yamamoto et al. [35], and Zhou et al. [38] among all the studies included. Both studies primarily had Asian patients. The patients enrolled in the Yamamoto study were of relatively good health, with no brain metastases and with a higher proportion of patients with an Eastern Cooperative Oncology Group performance status (ECOG-PS) score of 0, which may have contributed to the favorable PFS and OS data compared to other studies in advanced EGFR-positive NSCLC. In a study by Zhou et al., the erlotinib-plus-bevacizumab combination resulted in a significant PFS improvement in naive metastatic EGFR-mutated NSCLC with brain metastasis. The findings of this study showed that patients harboring exon 21 L858R derived more benefit from the addition of bevacizumab than from erlotinib alone. Overall, the entire study population and subgroups experienced improved PFS with no OS benefits and an improvement in ORR with combination therapy. The results of our meta-analysis were consistent with the above-mentioned data, suggesting that the treatment effect was quantitative rather than qualitative, favoring the same treatment with different effect sizes for treatment outcomes. This improved PFS value using erlotinib-plus-bevacizumab across all studies and among subgroups could be due to the change in tumor vessel physiology because of bevacizumab, thus resulting in an increased intra-tumoral uptake of drugs and thereby improving the drug delivery [39]. In addition, the effective blockade of angiogenesis signaling through the VEGF receptor and EGFR signaling pathways could have been useful in controlling the tumor growth [40]. Our findings appear valuable when compared to a study by Kenmotsu et al. [15], where an Osimertinib-plus-bevacizumab combination failed to exhibit efficacy against Osimertinib monotherapy in improving PFS in patients with non-squamous NSCLC harboring EGFR mutations [15]. Similarly, in another randomized clinical trial by Akamatsu et al. [16], Osimertinib-plus-bevacizumab combination failed to show improvement of PFS in patients with advanced lung adenocarcinoma with EGFR T790M mutation when compared with Osimertinib alone [16]. In addition, in a study by Soo et al. [17], no difference in PFS was observed between Osimertinib plus bevacizumab and Osimertinib alone, suggesting erlotinib plus bevacizumab elicited better PFS compared to erlotinib alone.

Although a significant improvement in PFS was demonstrated with combination therapy in the studies we analyzed, it did not result in a statistically significant benefit in OS across all trials and with subgroups. A study by Piccirillo et al. [33] showed a somewhat higher OS, although it was statistically non-significant. This was the only study conducted in Europe where the prevalence of EGFR mutations was low. A lack of OS benefits could have resulted from the availability of many treatment options for NSCLC, and first-line treatment outcomes on OS can be influenced by subsequent treatment. More patients received Osimertinib as a subsequent treatment in the erlotinib group than in the combination group, in the study by Zhou et al. [37].

The OS is influenced by a variety of factors, such as the impact of post-discontinuation therapy or follow-up treatment after drug resistance to the therapy, if patients suffered any emergencies, patients’ physical health, the line of treatment and the type of analysis conducted. In a randomized controlled trial, increased survival time is clearly an implicit goal, although it is difficult to establish when treatment change at the onset of new progressive events is a standard clinical practice. In cases where expected survival time may be very short, especially in the metastatic setting, the sequential use of multiple drug classes is very common. Thus, the use of an intention-to-treat approach has demonstrated an unequivocal survival benefit in a few studies. In addition, PFS was a primary outcome in a study by Saito, Seto, Zhou, Lee et al., rather than the OS used in all other included studies. Both PFS and OS are related, as OS encompasses both PFS and post-progression survival, but their relationship is not always linear in certain conditions like sarcoma, advanced breast cancer, prostate cancer, and NSCLC [41]. In the context of biologic and targeted therapies, the relationship between PFS and OS becomes even more complex [42]. The lack of statistical significance in OS could be due to various factors including differences in staging, tumor size, metastasis, patients receiving multiple lines of post-discontinuation therapy, the open-label design of the RCTs resulting in subjectivity regarding relevant treatments, and the type of analysis used (pre or post hoc). Additionally, a long post-progression survival (PPS) can reduce the statistical power to detect improvements in OS. Nonetheless, the absence of statistical significance in OS should not be taken as an indication of a lack of improvement in OS [36].

Similarly, no significant differences were observed for ORR between the two groups among studies included in this analysis, except in two studies where improvement in the ORR was observed with erlotinib plus bevacizumab [30,33]. The ORR was far better in a study by Herbst et al. [30] compared to all the other studies included. This could have been the result of having patients with ECOG scores of 2 or lower in this study. Similarly, the higher sensitivity of NSCLC to EGFT-TKIs, and requiring larger study populations due to high ORRs among the erlotinib-alone group, could be other possible reasons for a lack of improvement in ORR observed with the combination therapy. Overall, the ORR was insignificant between the erlotinib-plus-bevacizumab combination and erlotinib monotherapy group, and among subgroups. ORR is a tumor-based-assessment end point, and atypical responses observed with biologics or immunotherapy such as bevacizumab cannot be always captured with the assessment criteria used [43]. There may be chances of variability in the measurements of tumor size and heterogeneity both within a lesion and among different tumor lesions in a patient. Tumor shrinkage is not always symmetrical; they may exhibit heterogenous changes as a result of necrosis, fibrosis or intra-tumoral hemorrhage, especially with the use of biologic agents [44].

Notably, patients treated with the combination therapy showed significantly increased risk of grade 3 and higher AEs, in addition to risk of hypertension and proteinuria. These results are consistent with the findings of previous meta-analyses [21,22,23] and studies included in this meta-analysis. A greater number of patients suffered AEs of grade 3 and above, hypertension and proteinuria in a study by Seto et al., among the studies included. More patients suffered from AEs of grade 3 and above (95%), hypertension (60%) and proteinuria (8%) in the erlotinib-plus-bevacizumab combination therapy group compared to 31%, 10% and 0%, respectively, in the erlotinib alone group in this study. This could be attributed to the study’s longer median duration (16 cycles). The higher incidence of hypertension may be the result of the grading definition used in this study. The Common Terminology Criteria for Adverse Events (CTC-AE) version 4.03 was used to grade AEs in this study, rather than the CTC-AE version 3. This difference in the grading could have resulted in the higher incidence of AEs. Despite the higher incidence of hypertension in this study, bevacizumab administration was discontinued in only two (3%) patients. The reasons for treatment-related adverse events associated with treatment discontinuation because of skin rash or diarrhea were minimal in the included studies. Moreover, these were deemed non-serious and reversible. Hypertension is the most common AE associated with bevacizumab treatment. The exact reason for the increase of blood pressure associated with bevacizumab treatment is currently not known. It was hypothesized that VEGF inhibition may be responsible for reducing the nitric oxide synthase activity, thus affecting vascular smooth muscle cells and renal sodium elimination, resulting in increased arterial blood pressure, or this may be a result of the decreased micro-vessel perfusion and microvascular density resulting from VEGF blockade leading to increased peripheral resistance [45]. Hypertension associated with bevacizumab can be managed by following the general principles of hypertension management. It is central for healthcare professionals to evaluate the cardiovascular risk of each patient on bevacizumab and to maintain their blood pressure within the normal range [46]. Proteinuria commonly occurs in concert with hypertension. Evidence from large studies suggests that even mild increases in blood pressure in the long term have a strong independent risk of causing end-stage renal disease [47]. Although the AEs discussed are manageable, the effective management of AEs is crucial in achieving optimal clinical benefits and should be considered as an integral part of the overall treatment strategy for patients with advanced NSCLC.

We performed subgroup analyses to investigate the impact of the geographical region on treatment outcomes, since studies included both Asian and Caucasian populations. Although the geographic region did not modify the effect of the erlotinib-plus-bevacizumab combination compared to erlotinib monotherapy, the pooled effect estimates of PFS for the Asia subgroup favored the erlotinib-plus-bevacizumab combination more than the US/Europe subgroup. Moreover, this subgroup analysis showed no heterogeneity among the studies included. This can be attributed to the fact that EGFR mutation is the most common driver mutation gene in Asia, which accounts for approximately 50–60% of patients with lung adenocarcinoma compared to 10–20% in Caucasian patients [8]. Exon 19 deletions constitute 44.8% of EGFR mutations among East Asian patients [48]. Studies included in the Asia subgroup predominantly had patients with the exon-19-deletions EGFR mutation.

Studies included in this current meta-analysis had several limitations, including a small sample size, being open label [31,32,36,37,38], a lack of power to conduct sub-group analyses, immature data to examine OS [32,36], a failure to account for crossover effects of a potentially active therapy on overall survival [30], the use of Osimertinib as a subsequent treatment [37], variability in the use of the ECOG-PS status for patient enrollment, and a lack of diversity in the patient population, as most studies were conducted in an Asian population.

The efficacy of erlotinib plus bevacizumab compared to erlotinib monotherapy in EGFR-positive NSCLC patients was investigated in two other meta-analyses. One included six RCTs [23], while the other included observational studies along with RCTs [49]. In contrast, our meta-analysis not only exclusively included RCTs, but also incorporated three additional trials. It also had nearly an equal distribution of studies conducted in Asia and US/Europe. Therefore, our meta-analysis provides more precise and valid comparisons, and presents a comprehensive evaluation of the efficacy and safety of erlotinib-plus-bevacizumab combination compared to erlotinib monotherapy in patients with advanced EGFR-positive NSCLC. Another meta-analysis reported an improvement in PFS for patients treated with first-line angiogenesis inhibitors in combination with erlotinib, compared to erlotinib monotherapy [50]. Nonetheless, this meta-analysis included two distinct angiogenesis inhibitors, namely bevacizumab and ramucirumab. This may have introduced potential bias in the comparison between erlotinib-plus-bevacizumab combination and erlotinib monotherapy due to the differing therapeutic efficacies of the two drugs.

The current meta-analysis possessed certain potential limitations that must be considered. First, our quantitative synthesis included nine studies and a few of them had small sample sizes. Secondly, this meta-analysis was conducted at the trial level, and not at the individual-patient level. Thirdly, the dissimilarities in statistical quality and follow-up duration may have led to heterogeneity. We could not perform sensitivity analysis due to the small number of eligible studies and the fact that most of them were open-label. Consequently, patient co-morbidities, potential prognostic factors, and the extent of disease and confounding variables were not examined in this study. Furthermore, the lack of data on ctDNA for KDR/p53/other co-mutations in included studies limited the scope of our analyses. Therefore, future research should focus on high-quality studies that comprehensively evaluate the clinical features of patients, leading to more standardized and accurate conclusions. Despite these limitations, the rigorous study selection process used in this study allowed for a comprehensive assessment of the available evidence on the topic of interest. The final set of articles included in the meta-analysis represented a moderate- to high-quality sample of the relevant literature.

The management of lung cancer has progressed from using cytotoxic therapies to targeted therapies that are specific to molecular subtypes in recent years. Despite the emergence of promising treatments for patients with EGFR-positive NSCLC, the most effective strategy for multiple treatment lines remains uncertain. Our meta-analysis provides important insights for clinical practices when considering targeted therapy in NSCLC patients. Firstly, our analysis provides the most current and comprehensive evidence supporting the value of the combination of erlotinib plus bevacizumab over erlotinib monotherapy in NSCLC patients. Secondly, our findings indicate that the combination of erlotinib and bevacizumab could be recommended as a first-line treatment for selected patients, particularly those with EGFR-positive mutations. Lastly, when assessing the therapeutic approach for multiple treatment lines, clinicians now have an evidence-based preference towards utilizing the erlotinib-plus-bevacizumab combination as a viable primary treatment option. This approach enables the preservation of alternative treatments for subsequent treatment lines. The erlotinib and bevacizumab combination can be considered as a first-line therapeutic option in EGFR-positive advanced NSCLC patients. Future studies should investigate such novel combinations to identify patients that will benefit from such combinations and who are most likely to experience disease progression with single-agent EGFR TKIs.

## 5. Conclusions

In conclusion, the findings of our meta-analysis indicate a significant improvement in PFS with erlotinib-and-bevacizumab combination therapy. This combination can serve as a viable treatment option for EGFR-positive advanced NSCLC patients. These findings are also important considering the lack of improvement seen in PFS with Osimertinib and bevacizumab combination in recent studies. Although non-serious and reversible, an increase in adverse events associated with the use of erlotinib and bevacizumab combination therapy deserves further attention from healthcare providers and their appropriate management. The findings of this current meta-analysis could be further validated by identifying specific sub-groups that may significantly benefit from this combination therapy. Future investigations can look to modify the combination therapy using different TKIs or combining it with chemotherapy to optimize treatment options for patients with advanced NSCLC and EGFR-positive mutations to provide them with greater treatment benefits.

## Figures and Tables

**Figure 1 diseases-11-00146-f001:**
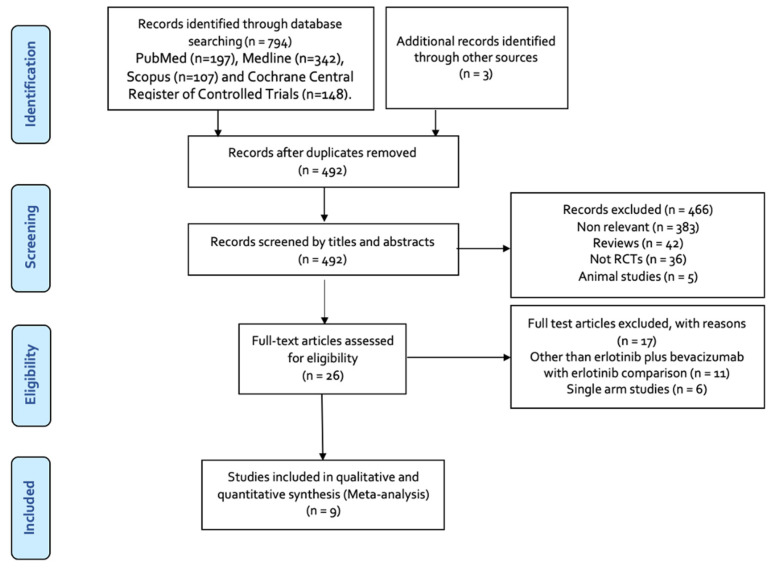
Flowchart of search and the eligible studies included in this meta-analysis.

**Figure 2 diseases-11-00146-f002:**
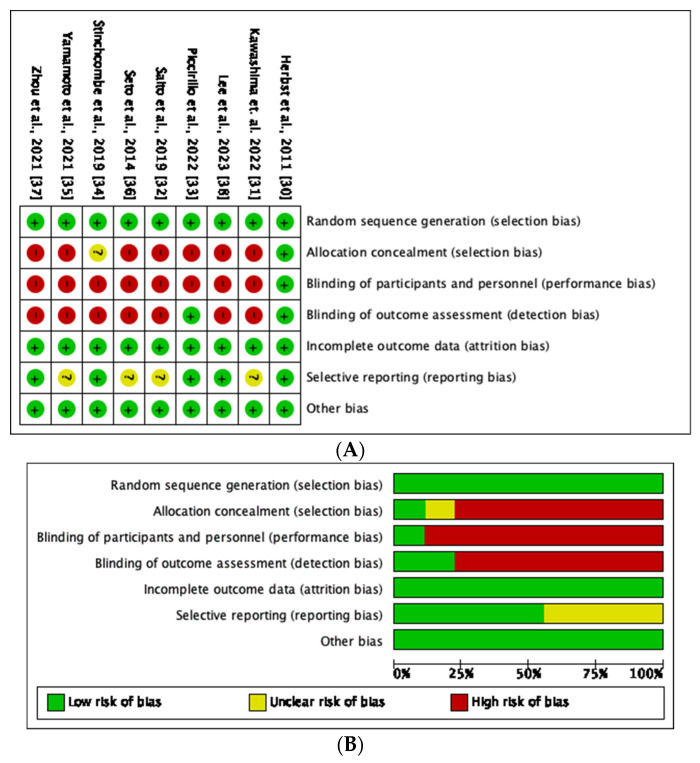
(**A**) Graphical representation of the risk of bias assessment. (**B**) Summary of the risk of bias assessment [30,31,32,33,34,35,36,37,38].

**Figure 3 diseases-11-00146-f003:**
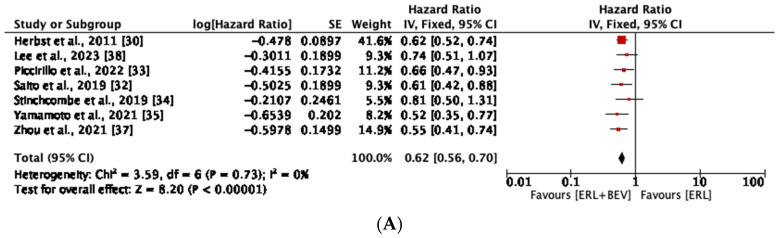
(**A**) Meta-analysis of progression-free survival (PFS) in entire study population with NSCLC receiving erlotinib-plus-bevacizumab combination and erlotinib monotherapy. (**B**) Meta-analysis of progression free survival (PFS) in subset of study population with NSCLC receiving erlotinib plus bevacizumab combination and erlotinib monotherapy based on geographic region [30,32,33,34,35,37,38].

**Figure 4 diseases-11-00146-f004:**
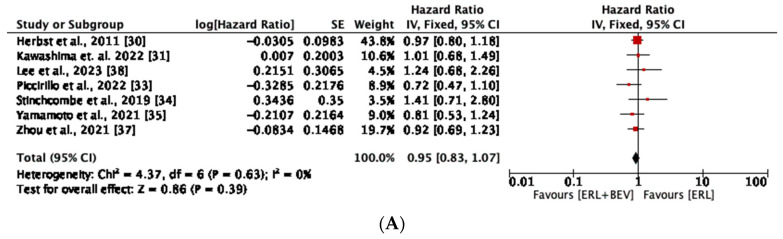
(**A**) Meta-analysis of overall survival (OS) in entire study population with NSCLC receiving erlotinib plus bevacizumab combination and erlotinib monotherapy. (**B**) Meta-analysis of overall survival (OS) in subset of study population with NSCLC receiving erlotinib-plus-bevacizumab combination and erlotinib monotherapy based on geographic region [30,31,33,34,35,37,38].

**Figure 5 diseases-11-00146-f005:**
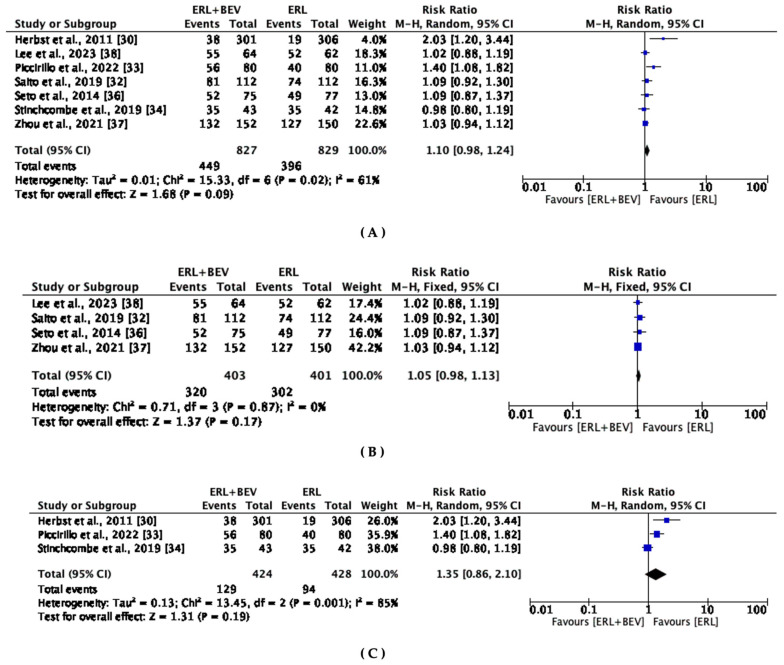
Meta-analyses of objective response rate (ORR) in (**A**) entire study population with NSCLC receiving erlotinib-plus-bevacizumab combination and erlotinib monotherapy, (**B**) studies in Asia, and (**C**) studies in US/Europe [30,32,33,34,36,37,38].

**Table 1 diseases-11-00146-t001:** Characteristics of the included studies.

Author, Year	Trial Name	Patients (N)	Phase	Study Region	Clinical Stage	Line of Treatment	Patients in (ERL + BEV) Group (n)	Patients in (ERL) Group (n)	Age (yrs.) (Median)	Female (%)	Outcome
Herbst et al., 2011 [30]	BeTa	636	III	US	I–IV	Second line	319	317	65	46	PFS, OS, ORR
Kawashima et. al., 2022 [31] ^(a)^	NEJ026	224	III	Japan	IIIb–IV	First-line	112	112	67	64	OS
Saito et al., 2019 [32] ^(a)^	NEJ026	224	III	Japan	IIIb–IV	First-line	112	112	67	64	PFS, ORR
Piccirillo et al., 2022 [33]	BEVERLY	160	III	Italy	IV	First-line	80	80	67	62.5	PFS, OS, ORR
Stinchcombe et al., 2019 [34]	NCT01532089	88	II	US	IV	First-line	43	45	63	70	PFS, OS, ORR
Yamamoto et al., 2021 [35] ^(b)^	JO25567	152	II	Japan	IIIb–IV	First-line	75	77	67	63	PFS, OS
Seto et al., 2014 [36] ^(b)^	JO25567	152	II	Japan	IIIb–IV	First-line	75	77	67	63	ORR
Zhou et al., 2021 [37]	ARTEMIS CTOG1509	311	III	China	IIIb–IV	First-line	157	154	58	62	PFS, OS, ORR
Lee et al., 2023 [38]	NCT03126799	127	II	Republic of Korea	IIIb–IV	First-line	64	63	63	66.1	PFS, OS, ORR

^(a)^ Dual publication of NEJ026 trial; ^(b)^ dual publication of JO25567 trial; ERL + BEV, erlotinib plus bevacizumab; ERL, erlotinib alone.

**Table 2 diseases-11-00146-t002:** Treatment outcomes (PFS, OS, ORR) in patients with NSCLC receiving erlotinib plus bevacizumab combination compared to erlotinib alone.

Author, Year	ORR: ERL + BEV (n)	ERL + BEV (N)	ORR: ERL (n)	ERL (N)	ORR—*p*-Value	PFS: ERL + BEV (Months)	PFS: ERL (Months)	PFS–HR, (95% CI)	OS: ERL + BEV (Months)	OS: ERL (Months)	OS–HR, (95% CI)
Herbst et al., 2011 [30]	38	301	19	306		3.4	1.7	0.62 (0.52, 0.75)	50.7	46.2	0.97 (0.80, 1.18)
Kawashima et. al., 2022 [31] ^(a)^											1.007 (0.68, 1.49)
Saito et al., 2019 [32] ^(a)^	81	112	74	112	0.31	16.9	13.3	0.61 (0.42, 0.88)			
Piccirillo et al., 2022 [33]	56	80	40	80	0.01	15.4	9.6	0.66 (0.47, 0.92)	33.3	22.8	0.72 (0.47, 1.10)
Stinchcombe et al., 2019 [34]	35	43	35	42	0.81	17.9	13.5	0.81 (0.50, 1.31)	32.4	50.6	1.41 (0.71, 2.81)
Yamamoto et al., 2021 [35] ^(b)^						16.4	9.8	0.52 (0.35, 0.76)	47	47.4	0.81 (0.53, 1.23)
Seto et al., 2014 [36] ^(b)^	52	75	49	77	0.49						
Zhou et al., 2021 [37]	132	152	127	150	0.56	17.9	11.2	0.55 (0.41, 0.73)	36.2	31.6	0.92 (0.69, 1.23)
Lee et al. 2023 [38]	55	64	52	62	0.476	17.5	12.4	0.74 (0.51, 1.08)			1.24 (0.68, 2.26)

^(a)^ Dual publication of NEJ026 trial; ^(b)^ dual publication of JO25567 trial; ORR, objective response rate; ERL+BEV, erlotinib plus bevacizumab; ERL, erlotinib alone; PFS, progression free survival; HR, hazard ratio; CI, confidence interval; OS, overall survival.

**Table 3 diseases-11-00146-t003:** Adverse events (AEs) in patients with NSCLC receiving erlotinib-plus-bevacizumab combination compared to erlotinib alone.

Author, Year	AE ≥ G3 (ERL + BEV)	AE ≥ G3 (ERL)	Skin Rash (ERL + BEV)	Rash (ERL)	HTN (ERL + BEV)	HTN (ERL)	Proteinuria (ERL + BEV)	Proteinuria (ERL)	Diarrhea (ERL + BEV)	Diarrhea (ERL)
Herbst et al., 2011 [30]	208 (66%)	165 (53%)	49 (16%)	19 (6%)	15 (5%)	4 (1%)				
Saito et al., 2019 [32]	98 (88%)	53 (46%)	23 (21%)	24 (21%)	26 (23%)	1 (1%)	8 (7%)	1 (1%)	6 (5%)	2 (2%)
Piccirillo et al., 2022 [33]	45 (56%)	39 (49%)	73 (92%)	70 (88%)	19 (24%)	4 (5%)	5 (6%)	1 (1%)	4 (5%)	3 (4%)
Stinchcombe et al., 2019 [34]	26 (60%)	15 (33%)	11 (26%)	7 (16%)	17 (40%)	9 (20%)	5 (12%)	0 (0%)	4 (9%)	6 (13%)
Seto et al., 2014 [36]	71 (95%)	24 (31%)	19 (25%)	15 (19%)	45 (60%)	8 (10%)	6 (8%)	0 (0%)	1 (1%)	1 (1%)
Zhou et al., 2021 [37]	63 (40%)	15 (9%)	8 (5%)	5 (3%)	29 (19%)	5 (3%)	11 (7%)	0 (0%)	4 (3%)	0 (0%)
Lee et al., 2023 [38]	29 (45%)	6 (9.5%)	11 (17.2%)	3 (4.8%)	9 (14.1%)	0 (0%)	5 (7.8%)	0 (0%)	4 (6.3%)	3 (4.8%)

ERL + BEV, erlotinib plus bevacizumab; ERL, erlotinib alone; AE, adverse events; HTN, hypertension.

**Table 4 diseases-11-00146-t004:** Relative risk of adverse events (AEs) in patients with advanced NSCLC treated with erlotinib-plus-bevacizumab combination compared to erlotinib alone.

Adverse Events	ERL + BEV Event/Total	ERL Event/Total	RR (95%CI)	*p*-Value	Heterogeneity		Model Type
					I^2^	*p*-Value	
Grade 3 AEs	540/844	317/844	2.09 (1.47, 2.97)	<0.00001	90%	<0.00001	Random effect
Skin rash	194/844	143/844	1.53 (0.92, 2.52)	0.10	84%	<0.00001	Random effect
Hypertension	160/844	31/844	5.15 (3.59, 7.39)	<0.00001	47%	0.08	Fixed effect
Diarrhea	23/531	15/531	1.53 (0.48, 2.86)	0.18	0%	0.56	Fixed effect
Proteinuria	46/531	2/531	12.03 (4.37, 33.17)	<0.00001	0%	0.94	Fixed effect

ERL + BEV, erlotinib plus bevacizumab; ERL, erlotinib alone; RR, relative risk; CI, confidence interval.

## Data Availability

Data included in this research study was abstracted from the published literature.

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
