# Peer review of "Comparing Efficacy of Erlotinib and Bevacizumab Combination with Erlotinib Monotherapy in Patients with Advanced Non-Small Cell Lung Cancer (NSCLC): A Systematic Review and Meta-Analysis"

_diseases, 2023, doi:10.3390/diseases11040146_

Round 1
Reviewer 1 Report
The authors provide a well written meta-analysis of erlotinib with bevacizumab. The re-emphasized PFS but not OS benefit has clinical utility in treating EGFR mutated NSCLC, in particular with the impact of bevacizumab with atezolizumab and chemotherapy as a salvage treatment approach. Unfortunately, none of the erlotinib-bevacizumab studies do not include ctDNA for KDR/p53/other co-mutations which just might idnetify a sub-group with OS benefit.
I do think two points for a minor revision would enhance the clarity of the manuscript.
1) Lines 41-43--The current standard of care in metastatic NSCLC without a driver mutation or fusion is immune checkpoint inhibitor-based treatment, not cytotoxic chemotherapy alone.
2) Although not directly part of the erlotinib-bevacizumab meta-analysis, a comment in the discussion regarding the osimertinib-bevacizumab data (Kenmotsu et al J Thorac Oncol 2022; Yu et al Akamatsu et al JAMA Oncol 20221; Soo et al Ann Oncol 2021) would provide a better context of clinically interpreting the meta-analysis data given the median OS benefit of osimertinib over erlotinib/gefitinib and no PFS benefit of bevacizumab with osimertinib.
Author Response
Dear Reviewer,
We sincerely thank you and appreciate your comments that helps us to improve our manuscript significantly. Appended below our point to point response to your valuable comments.
The authors provide a well written meta-analysis of erlotinib with bevacizumab. The re-emphasized PFS but not OS benefit has clinical utility in treating EGFR mutated NSCLC, in particular with the impact of bevacizumab with atezolizumab and chemotherapy as a salvage treatment approach. Unfortunately, none of the erlotinib-bevacizumab studies do not include ctDNA for KDR/p53/other co-mutations which just might identify a sub-group with OS benefit.
We appreciate reviewer’s compliments on our manuscript. Thank you for pointing out this important limitation. We concur with the reviewer’s comment. Studies included in this SR/meta-analysis did not include ctDNA for KDR/p53/other co-mutations thus limiting the scope of our analyses. We have acknowledged this as one of the limitations of our study.
I do think two points for a minor revision would enhance the clarity of the manuscript.
1) Lines 41-43--The current standard of care in metastatic NSCLC without a driver mutation or fusion is immune checkpoint inhibitor-based treatment, not cytotoxic chemotherapy alone.
We thank the reviewer for pointing this out. We have modified this sentence as suggested. The text now reads as: The immune checkpoint inhibitor-based treatment, not a cytotoxic chemotherapy alone is current standard of care in metastatic NSCLC without a driver mutation or fusion.
2) Although not directly part of the erlotinib-bevacizumab meta-analysis, a comment in the discussion regarding the osimertinib-bevacizumab data (Kenmotsu et al J Thorac Oncol 2022; Yu et al Akamatsu et al JAMA Oncol 20221; Soo et al Ann Oncol 2021) would provide a better context of clinically interpreting the meta-analysis data given the median OS benefit of osimertinib over erlotinib/gefitinib and no PFS benefit of bevacizumab with osimertinib.
We thank the reviewer for this suggestion and providing additional references. We have included these references and modified our discussion accordingly, which now reads as- Our study findings seem to be valuable when compared to a study by Kenmotsu and colleagues [15], where osimertinib plus bevacizumab combination failed to exhibit the efficacy against osimertinib monotherapy in improving the PFS in patients with non-squamous NSCLC harboring EGFR mutations [32]. Similarly, in another randomized clinical trial by Akamatsu et al. [16], osimertinib plus bevacizumab combination failed to show improvement of PFS in patients with advanced lung adenocarcinoma with EGFR T790M mutation when compared with osimertinib alone [33]. In addition, in a study by Soo and colleagues [17], no difference in PFS was observed between osimertinib plus bevacizumab and osimertinib alone suggesting erlotinib plus bevacizumab had better PFS compared to erlotinib alone.
Reviewer 2 Report
Thank you for the opportunity to review the manuscript titled "Comparing Efficacy of Erlotinib and Bevacizumab Combination with Erlotinib Monotherapy in patients with Advanced Non-Small Cell Lung Cancer (NSCLC): A Systematic Review and Meta-Analysis" (diseases-2483252). I find the subject of the article exciting; however, there are several areas where the text could be improved. I believe that with major revisions, this work has the potential to be accepted for publication. Please find my detailed comments below:
Title:
- In the title the “patients” should be changed with “Patients”.
1. Introduction:
- The sentence "In the United States, lung cancer is the second most common cancer in both men and women [1]." could be rephrased for better clarity, e.g., "Lung cancer is the second most common cancer in both men and women in the United States [1]."
- The reference citation for the statistic on the prevalence of EGFR mutations in metastatic NSCLC should be included within the sentence.
- The reference citations for the studies that have shown the efficacy of erlotinib in improving PFS and OS in patients with EGFR-positive NSCLC should be included within the sentence.
- The reference citations for the studies that have shown the efficacy of bevacizumab in improving OS in advanced NSCLC patients should be included within the sentence.
- The statement "However, these results were inconclusive due to lack of a comparison group in this study." could be clarified by mentioning that the study did not include a control group for comparison.
- The previous conflicts in clinical trial results, underpowering, and reporting/publication bias should be supported with specific reference citations.
- The mentioning of previous meta-analyses investigating EGFR-TKIs in combination with VEGFR-TKI should be supported with specific reference citations.
- The sentence "One such meta-analysis investigated erlotinib plus bevacizumab combination compared to erlotinib monotherapy in the treatment of NSCLC; however, it has reported on only six randomized clinical trials (RCTs), primarily from Asia [16]." could be rephrased for better clarity, e.g., "However, a previously conducted meta-analysis on this topic only included six RCTs, primarily from Asia [16]."
- The statement about utilizing a PICO framework could be rephrased for clarity, e.g., "Using a PICO framework (Patient, Intervention, Comparator, Outcomes), we conducted a meta-analysis to investigate whether the combination of erlotinib and bevacizumab is associated with improved OS, PFS, and ORR compared to erlotinib in NSCLC patients."
- The mention of performing subgroup analyses to assess the impact of geographic regions should be clarified by mentioning the specific regions included in the analysis.
2. Materials and Methods
- In the sentence "To identify relevant articles, we systematically searched several databases, including PubMed, Medline, Scopus, Cochrane Central Register of Controlled Trials, American Society of Clinical Oncology (ASCO), International Association for the Study of Lung Cancer (IASLC), European Society of Medical Oncology (ESMO), and clinicaltrials.gov," consider adding a semicolon or bullet points to separate the list of databases for clarity.
- The sentence "The last search was conducted in March 2023" should be revised as it is not possible to conduct a search in the future.
- In the sentence "Additionally, bibliographies of relevant articles were searched for additional publications," consider rephrasing to "Additionally, the bibliographies of relevant articles were searched for additional publications."
- In the sentence "The literature search was performed independently by two researchers (PS and SK)," specify whether the researchers searched the literature individually or together.
- In the sentence "If multiple articles covered the same study population, study with the most recent treatment outcome data was utilized," clarify what the researchers mean by "utilized." Do they mean that they selected the most recent study or used the data from the most recent study?
- In the sentence "We included randomized clinical trials only that met the following conditions," consider rephrasing as "We only included randomized clinical trials that met the following conditions."
- In the sentence "Studies that did not fulfill these criteria, such as single-arm studies, animal experimentation, case reports, systematic review, and meta-analysis, and those published in languages other than English were excluded," consider separating the list of excluded study types for clarity.
- In the sentence "The data extracted from the studies included the first author, year of publication, region, study type, number of patients in each group, type of EFGR mutation, therapeutic regimen, survival outcomes, and information on adverse drug events," check if "EFGR" should be spelled as "EGFR."
- In the sentence "The methodological quality of the studies included in the investigation was evaluated using the Cochrane Risk of Bias tool [18], which assesses quality on seven domains," clarify what the seven domains are or provide a reference to the Cochrane Risk of Bias tool.
- In the sentence "We estimated the pooled hazard ratios (HR) of OS and PFS with a 95% confidence interval (CI)," consider expanding the abbreviations "OS" and "PFS" for clarity.
- In the sentence "Subgroup analyses were performed to compare treatment effect according to geographic location," consider specifying which geographic locations were included in the subgroup analyses.
- In the sentence "Publication bias was assessed using the funnel plot and Egger's test," explain what Egger's test is and how it is used to assess publication bias.
Consider adding references for the statistical analysis methods and software used. - Lastly, consider rephrasing the sentence "A p-value of <0.05 was considered statistically significant" to "Statistical significance was defined as a p-value of <0.05."
3.1. Results of the literature search
- The study selection process is visually presented in Figure 1.
- Initially, 797 records were identified through the search.
- After removing duplicates, 305 records were excluded.
- 17 articles were excluded during screening for eligibility.
- This resulted in a final set of nine studies [21-29] that met the inclusion criteria and were included in the meta-analysis.
3.2. Characteristics of the included studies
- able 1 summarizes the characteristics of the nine included studies.
- The meta-analysis included a total of 1,698 patients from these nine studies.
- The NEJ026 study [22-23] and the JO25567 study [26-27] had dual publications included in the analysis.
- The erlotinib plus bevacizumab group consisted of 850 NSCLC cases, while the erlotinib group included 848 cases.
- Among the nine included RCTs, five were phase-3 trials and four were phase-2 trials.
- The earliest study was published in 2011 and the most recent in 2023.
- The sample size in these studies ranged from 88 to 363 patients.
- The median age of the patients ranged from 58 to 67 years, and the proportion of female patients ranged from 46% to 70%.
- Seven studies included NSCLC patients with L858R or del19 EGFR mutation.
- Patients in all studies received erlotinib plus bevacizumab combination as a first line of treatment, except for one study.
3.3. Risk of bias and quality assessment
- Figure 2A provides a summary of the risk of bias for each included study.
- Majority (8) of the studies were open-label.
- Only one study had adequate concealment of allocation.
- All studies were free of incomplete outcome data and other bias.
- The studies included in this meta-analysis were found to be of moderate to high quality (Figure 2B).
3.4. Progression free survival
- Except for two studies, all others reported statistically significant improvement in PFS.
- The median PFS ranged from 3.4 to 17.9 months in the combination group compared to 1.7 to 13.5 months in the erlotinib monotherapy group (Table 2).
3.5. Overall survival
- All included studies reported no statistically significant improvement in OS.
- The median OS ranged between 9.3 to 50.7 months in the combination group compared to 9.2 to 50.6 months in the erlotinib monotherapy group (Table 2).
- The meta-analysis showed that erlotinib plus bevacizumab combination did not significantly prolong OS compared to erlotinib monotherapy in the entire study population (HR, 0.95; 95% CI: 0.83-1.07, p=0.39) (Figure 4A).
- Subgroup analyses also did not show significant improvement in OS with the combination therapy (Figure 4B).
- No heterogeneity was observed across all the analyses.
- Geographic region did not significantly modify the effect of the combination therapy compared to monotherapy on the test for subgroup differences (p=0.99).
3.6. Objective response rate
- Except for two studies, all others reported no statistically significant improvement in ORR.
- The median ORR ranged from 12.6% to 86.8% in the combination group compared to 6.2% to 83.9% in the erlotinib monotherapy group (Table 2).
- The results indicated that erlotinib plus bevacizumab combination did not significantly improve ORR compared to erlotinib monotherapy in the entire study population (HR, 1.10; 95% CI: 0.98-1.24, p=0.09) (Fig. 5A).
- Subgroup analyses also did not show significant improvement in ORR with the combination therapy (Figures 5A, B, and C).
- Significant heterogeneity was observed across all analyses, except for the subgroup analysis of studies from Asia.
- Geographic region had no effect on the treatment outcome of the combination therapy compared to monotherapy (p=0.17).
3.7. Adverse events
- AEs associated with the use of erlotinib plus bevacizumab were compared to those associated with erlotinib monotherapy.
- Only grade 3 and higher AEs were analyzed (Table 3).
- The results of the meta-analysis showed that combination therapy was associated with a higher incidence of high-grade AEs compared to monotherapy.
- Specifically, patients treated with erlotinib plus bevacizumab had significantly increased risks of hypertension, proteinuria, and grade 3 and higher AEs.
- However, the risk of skin rash and diarrhea were comparable between the two groups.
3.8. Publication bias
- There was no evidence of apparent publication bias based on the assessment by funnel plot and Egger's test (p > 0.05).
4. Discussion
- The discussion section is well-written and provides a clear and concise overview of the findings and their implications.
- The use of abbreviations, such as PFS, OS, ORR, and AEs, is appropriate and helps in conveying the information more efficiently.
- The section could benefit from providing more specific information about the number of studies included in the meta-analysis and the total number of patients analyzed.
- It would be helpful to include the statistical methods used in the meta-analysis, such as the type of model for the pooled effect estimates and any sensitivity analyses performed.
- The section could provide more details about the specific AEs observed with the combination therapy and their clinical implications.
- The potential reasons for the lack of statistically significant improvement in OS could be expanded upon, such as discussing the impact of post-discontinuation therapy and the type of analysis used.
- The section could discuss any limitations or biases in the individual studies included in the meta-analysis, such as the open-label design or potential confounding variables.
- More details about the specific inclusion and exclusion criteria for the studies included in the meta-analysis could be provided.
- While the discussion provides valuable insights into the clinical practice, it would be useful to discuss any potential future research directions based on the findings.
- The section could benefit from a stronger conclusion summarizing the main findings and their implications for clinical practice.
5. Conclusions
- ”erlotinib and bevacizumab significantly increases PFS" - The word "increases" should be changed to "increase" to match the subject-verb agreement.
- "is a viable treatment option" - Consider adding "a" before "viable" to improve the sentence's clarity.
- "for patients with advanced NSCLC who have EGFR positive mutations" - Consider rephrasing this part to "for patients with EGFR-positive mutations in advanced NSCLC."
- "it does not appear to significantly improve OS and ORR" - Consider rephrasing this to "our analysis does not show significant improvements in OS and ORR."
- "Clinicians should also consider the potential for increased adverse events" - Consider rephrasing this to "Clinicians should also consider the potential increase in adverse events."
- "associated with the use of erlotinib plus bevacizumab" - Consider rephrasing this to "associated with using the combination of erlotinib and bevacizumab."
- "Additional research is needed to validate these results" - Consider adding "in order" before "to validate."
- "identify patient subgroups that may benefit" - Consider adding "that" before "may benefit."
- "this combination therapy" - Consider specifying "the combination therapy of erlotinib and bevacizumab."
- "determine optimal treatment strategies" - Consider specifying "the optimal treatment strategies."
- "for patients with EGFR positive advanced NSCLC" - Consider rephrasing this to "for patients with advanced NSCLC and EGFR-positive mutations."
Here are some useful references for the author's observation
https://www.sciencedirect.com/science/article/pii/S0378517322005828
https://www.ncbi.nlm.nih.gov/pmc/articles/PMC6934956/
English needed major revision.
Author Response
Dear Reviewer,
We sincerely thank you for your comprehensive and detailed review of our manuscript,
Your valuable comments helped us to improve our manuscript significantly.
Please refer to the attachment for our our point-to-point response to your comments. Hope this helps to satisfy your concerns.
Thank you.

Reviewer 3 Report
The manuscript titled “Comparing Efficacy of Erlotinib and Bevacizumab Combination with Erlotinib Monotherapy in patients with Advanced 3 Non-Small Cell Lung Cancer (NSCLC): A Systematic Review 4 and Meta-Analysis” provides an update on the current clinical treatment strategy for NSCLC.
1. The manuscript lacks current up to date information. More relevant and recent references are needed
2. There has been a lot of publications out there that discuss this treatment strategy. I believe this manuscript does not add much new information than theres already available
3. Font size for all tables needs to be increased
4. Fig 3-5 font size needs to be increased
5. Expand the adverse events section to include some examples to make this a comprehensive read
6. It is recommended to provide examples for each section that is discussed- similar to comment 5
Author Response
Dear Reviewer,
We sincerely thank you for your comprehensive review of our manuscript,
Your valuable comments helped us to improve our manuscript significantly.
Appended below our point-to-point response to your comments. Hope this helps to satisfy your concerns.
The manuscript titled “Comparing Efficacy of Erlotinib and Bevacizumab Combination with Erlotinib Monotherapy in patients with Advanced 3 Non-Small Cell Lung Cancer (NSCLC): A Systematic Review 4 and Meta-Analysis” provides an update on the current clinical treatment strategy for NSCLC.
- The manuscript lacks current up to date information. More relevant and recent references are needed
We thank you for the above suggestion. We have now included a few more relevant and recent references in the manuscript. The text now reads as:
The erlotinib and bevacizumab combination in one phase III clinical study showed greater PFS in unselected populations of patients with NSCLC [14]. On the other hand, studies that compared Osimertinib plus bevacizumab against Osimertinib monotherapy failed to demonstrate improved PFS in patients with NSCLC harboring EGFR mutations [15-17].
Our study findings seem to be valuable when compared to a study by Kenmotsu and colleagues [15], where osimertinib plus bevacizumab combination failed to exhibit the efficacy against osimertinib monotherapy in improving the PFS in patients with non-squamous NSCLC harboring EGFR mutations [32]. Similarly, in another randomized clinical trial by Akamatsu et al. [16], osimertinib plus bevacizumab combination failed to show improvement of PFS in patients with advanced lung adenocarcinoma with EGFR T790M mutation when compared with osimertinib alone [33]. In addition, in a study by Soo and colleagues [17], no difference in PFS was observed between osimertinib plus bevacizumab and osimertinib alone suggesting erlotinib plus bevacizumab had better PFS compared to erlotinib alone.
The exact reason for the increase of blood pressure associated with bevacizumab treatment is currently not known. It was hypothesized that VEGF inhibition may be responsible for reducing the nitric oxide synthase activity, thus affecting vascular smooth muscle cells and renal sodium elimination, resulting in increased arterial blood pressure or it may be a result of decreased micro-vessel perfusion and microvascular density resulting from VEGF blockade leading to increased peripheral resistance [45]. Hypertension associated with bevacizumab can be managed by following the general principles of hypertension management. It is central for healthcare professionals to evaluate the cardiovascular risk of each patient on bevacizumab and to maintain their blood pressure within the normal range [46]. Proteinuria commonly occurs in concert with hypertension. Evidence from large studies suggests that even mild increases in blood pressure in the long term, have a strong independent risk of developing end stage renal disease [47].
- There has been a lot of publications out there that discuss this treatment strategy. I believe this manuscript does not add much new information than theres already available.
We thank you for this valuable suggestion. Although, it appears on a face value that there are lot of publications discussing erlotinib + bevacizumab combination strategy, we were able to find only limited number of studies that met our inclusion criteria of studies comparing erlotinib + bevacizumab combination with erlotinib alone. Unfortunately, we could not include other potential publications that compared bevacizumab in combination with other EGFR-TKIs, such as Gefitinib, Sunitinib, Osimertinib etc. since they did not meet our inclusion criteria and such comparison wasn't the objective of our study.
- Font size for all tables needs to be increased
Thank you for the suggestion. We have increased the font size for tables and figures and included them as an independent file. (This was due to the limitation posed by template provided by the journal for formatting of the manuscript. The fixed page width of the templet limited our ability of fitting large tables or figures on one page, and forced us to use smaller font size)
- Fig 3-5 font size needs to be increased
Please refer to the response above.
- Expand the adverse events section to include some examples to make this a comprehensive read
Thank you for the suggestion. We have now expanded this section to include more examples as suggested. This section now reads as- Greater number of patients suffered grade 3 and above AEs, hypertension, and proteinuria in a study by Seto and colleagues among the studies included. More patients suffered grade 3 and above AEs (95%), hypertension (60%) and proteinuria (8%) in erlotinib plus bevacizumab combination therapy group compared to 31%, 10% and 0%, respectively in erlotinib alone group in this study. This could be attributed to the study’s longer median study duration (16 cycles). The higher incidence of hypertension may be the result of grading definition used in this study. The CTC-AE version 4.03 was used to grade AEs in this study rather than the CTC-AE version 3. This difference in the grading could have resulted in higher incidence of AEs. Despite the higher incidence of hyper-tension in this study, bevacizumab administration was discontinued only in two (3%) patients. The reasons for treatment related adverse event associated with treatment discontinuation because of skin rash or diarrhea were minimal in included studies. Moreover, these were deemed non-serious and reversible. Hypertension is the most common AE associated with bevacizumab treatment. The exact reason for the increase of blood pressure associated with bevacizumab treatment is currently not known. It was hypothesized that VEGF inhibition may be responsible for reducing the nitric oxide syn-thase activity, thus affecting vascular smooth muscle cells and renal sodium elimination, resulting in increased arterial blood pressure or it may be a result of decreased mi-cro-vessel perfusion and microvascular density resulting from VEGF blockade leading to increased peripheral resistance [45]. Hypertension associated with bevacizumab can be managed by following the general principles of hypertension management. It is central for healthcare professionals to evaluate the cardiovascular risk of each patient on bevacizumab and to maintain their blood pressure within the normal range [46]. Proteinuria commonly occurs in concert with hypertension. Evidence from large studies suggests that even mild increases in blood pressure in the long term, have a strong independent risk of developing end stage renal disease [47]. AEs discussed above are although manageable, effective management of AEs is crucial in achieving optimal clinical benefits and should be considered as an integral part of the overall treatment strategy for patients with advanced NSCLC.
- It is recommended to provide examples for each section that is discussed- similar to comment 5.
Thank you for the suggestion. We have now modified our discussion to include more examples as suggested. This now reads as - The greater PFS among erlotinib and bevacizumab combination therapy group was compared to erlotinib alone group observed in a study by Yamamoto et al. [36], and Zhou et al. [38] among all the studies included. Both studies primarily had Asian patients. The patients enrolled in Yamamoto study were of relatively good health with no brain metastases and with a higher proportion of patients with an Eastern Cooperative Oncology Group performance status (ECOG-PS) score of 0, which may have contributed to the favorable PFS and OS data compared to other studies in advanced EGFR positive NSCLC. In a study by Zhou and colleagues, erlotinib plus bevacizumab combination resulted in significant PFS improvement in naive metastatic EGFR mutated NSCLC with brain metastasis. Finding of this study showed patients harboring exon 21 L858R derived more benefit from the addition of bevacizumab than from erlotinib alone.
Our study findings seem to be valuable when compared to a study by Kenmotsu and colleagues [15], where osimertinib plus bevacizumab combination failed to exhibit the efficacy against osimertinib monotherapy in improving the PFS in patients with non-squamous NSCLC harboring EGFR mutations [32]. Similarly, in another randomized clinical trial by Akamatsu et al. [16], osimertinib plus bevacizumab combination failed to show improvement of PFS in patients with advanced lung adenocarcinoma with EGFR T790M mutation when compared with osimertinib alone [33]. In addition, in a study by Soo and colleagues [17], no difference in PFS was observed between osimertinib plus bevacizumab and osimertinib alone suggesting erlotinib plus bevacizumab had better PFS compared to erlotinib alone.
Although a significant improvement in PFS was demonstrated with combination therapy in the studies we analyzed, it did not result in a statistically significant benefit in OS across all trials and with subgroups. A study by Piccirillo and colleagues [33] showed somewhat higher OS, although it was non-significant statistically. This was the only study conducted in Europe where prevalence of EGFR mutations is low. Lack of OS benefits could have been resulted because of availability of many treatment options for NSCLC and first-line treatment outcome on OS can be influenced by subsequent treatment. More patients received Osimertinib as a subsequent treatment in the erlotinib group than in the combination group in a study by Zhou et al.[37].
The ORR was far better in a study by Herbst and colleagues [30] among all studies included. This could have been resulted from having patients with ECOG scores of 2 or lower in this study. Similarly, higher sensitivity of NSCLC to EGFT-TKIs and requiring larger study population due to high ORR among erlotinib alone group could be possible reasons for lack of improvement in ORR observed with the combination therapy.
Round 2
Reviewer 2 Report
Acceptable for publication.